# Progressive State Space Disaggregation for Infinite Horizon Dynamic Programming

**Primary Keywords:** *(4) Theory; (6) Temporal Planning;*

## Abstract

High dimensionality of model-based Reinforcement Learning and Markov Decision Processes can be reduced using abstractions of the state and action spaces. Although hierarchical learning and state abstraction methods have been explored over the past decades, explicit methods to build useful abstractions of models are rarely provided. In this work, we provide a new state abstraction method for solving infinite horizon problems in the discounted and total settings. Our approach is to progressively disaggregate abstract regions by iteratively slicing aggregations of states relatively to a value function. The distinguishing feature of our method, in contrast to previous approximations of the Bellman operator, is the disaggregation of regions during value function iterations (or policy evaluation steps). The objective is to find a more efficient aggregation that reduces the error on each piece of the partition. We provide a proof of convergence for this algorithm without making any assumptions about the structure of the problem. We also show that this process decreases the computational complexity of the Bellman operator iteration and provides useful abstractions. We then plug this state space disaggregation process in classical Dynamic Programming algorithms namely Approximate Value Iteration, Q-Value Iteration and Policy Iteration. Finally, we conduct a numerical comparison which shows that our algorithm is faster than both traditional dynamic programming approach and recent aggregative methods that use a fixed number of adaptive partitions.

## Introduction

The Markov Decision Process (MDP) serves as a comprehensive framework for addressing stochastic dynamic control problems. Within this framework, the environment undergoes stochastic evolution influenced by the actions of an agent. The primary objective is to optimize expected gains through strategic decision-making (Puterman 2014). The overarching objective is to identify the optimal sequence of actions, referred to as a policy, that maximizes the overall return. This pursuit extends to a diverse array of problem domains, as highlighted in a recent overview (Boucherie and Van Dijk 2017). These encompass challenges in inventory control, energy management, network optimization involving queues, and navigating stochastic shortest paths in robot exploration. Achieving near-optimal control is crucial, necessitating precise solutions to effectively address these problems.

The curse of dimensionality is a well-documented challenge in the Markov Decision Process (MDP) framework, particularly when dealing with large state spaces. To overcome this, various strategies decompose complex MDPs into more manageable counterparts. Notably, Factored MDPs (Guestrin et al. 2003) represent states as dynamic feature vectors, using Dynamic Bayesian Networks for compact representation and efficient computation. Another recent approach, Reduced-Rank MDPs (Siddiqi, Boots, and Gordon 2010), expresses transition probabilities as scalar products of continuous functions, offering an effective dimensionality reduction technique. A general and promising method for MDP approximation is the Hierarchical Solution (Hengst 2012), which considers either temporal abstractions for actions persisting over time (Sutton, Precup, and Singh 1999), or state abstractions by aggregating states into meaningful regions (Li, Walsh, and Littman 2006), enhancing efficiency in handling complex MDPs.

In the context of state aggregation, the key challenge lies in determining the optimal grouping of different states and evaluating the quality of this aggregation. The selection of merging criteria becomes pivotal, and various criteria have been proposed in the literature. For instance, (Dean and Givan 1997) employs bisimulation for state grouping, (Singh, Jaakkola, and Jordan 1994) introduces a soft aggregation where states have probabilities of belonging to an aggregated region, (Ferrer-Mestres et al. 2020) restricts the number of regions, and (Abel, Hershkowitz, and Littman 2016) provides a compilation of several aggregation criteria. Assessing the quality of aggregation involves leveraging results from approximated dynamic programming and stochastic optimization, as discussed in (Tsitsiklis and Van Roy 1996) and further explored in (Abel, Hershkowitz, and Littman 2016; Abel 2019).

The incorporation of the aggregation process into solution algorithms, such as Modified Policy Iteration (MPI) or Value Iteration (VI), serves to enhance computational efficiency. In the realm of MPI, (Bertsekas, Castanon et al. 1988) employs aggregation based on the Bellman Residual to accelerate the process. More recently, (Chen et al. 2022) applies aggregation to VI by grouping states with similar values, demonstrating a contemporary approach to improving computational speed.

We introduce a class of iterative aggregation algorithms

for solving infinite horizon problems with both expected discounted and expected total criteria. Our approach integrates state abstraction with Approximate Value Iteration, Q-Value, and Policy Iteration algorithms. The iterative process begins by consolidating the entire state space into a single region. Subsequently, at each step, the current regions are subdivided based on the states' current values. The process concludes when each region groups states with similar values and when the current value function approximates the optimal one. A key innovation is the progressive aggregation along iteration steps, gathering states with similar evolution under the Bellman operator application. This refined state abstraction enhances algorithm efficiency. We provide a convergence proof without assumptions about the problem's structure, demonstrating decreased computational complexity and valuable abstractions. Numerical comparisons across diverse models underscore the algorithm's favorable standing in the Markov Decision Process literature, particularly in comparison to other abstraction algorithms.

The structure of the article unfolds as follows: initially, we establish a connection between approximate Bellman operators with State Aggregation and the Bellman operator of the abstract MDP. Subsequently, we articulate an error bound for the optimal value function, contingent on the quality of the aggregation employed. Following this theoretical foundation, we introduce our algorithms. Lastly, we assess the efficacy of our method through benchmarking on classical models, showcasing its efficiency in comparison to alternative approaches.

## Problem Setup

Our approach is grounded in Markov Decision Processes, a well-documented field. We clarify notations, outline Dynamic Programming methods for model resolution, and integrate recent advancements in State Abstraction and Approximate Dynamic Programming.

**Markov Decision Processes**   Markov Decision Processes provide a framework for decision-making optimization (Puterman 2014). Formally, a finite MDP is specified as a tuple $\langle \mathcal{S}, \mathcal{A}, T, R, \gamma \rangle$, where $\mathcal{S}$ is the set of possible states, $\mathcal{A}$ is the set of actions that the agent can select, $T(s, a, s') \in [0, 1]$ is the environment transition probability from $s$ to $s'$ under action $a$ and $R(s, a) \in \mathbb{R}$ describes the reward received by the agent in $s$ triggering action $a$. Finally, we consider bounded rewards and a discount factor $\gamma \in (0, 1]$ to weight the incoming reward priority.

The objective is to maximize the expected sum of discounted immediate rewards in the upcoming trajectory of states for an infinite horizon. The researched solution is a deterministic policy $\pi : \mathcal{S} \mapsto \mathcal{A}$ that can decide which action to select when in state $s \in \mathcal{S}$. For a given policy $\pi$, it is thus possible to define the value function that gives a value to each state. It is defined as the expected return applying the policy $\pi$ and we have $\forall s \in \mathcal{S}$:

$$V^{\pi}(s) = \mathop{\mathbb{E}}_{s_{t+1} \sim T(s_t, a_t, \cdot)} \left[ \sum_{t=0}^{\infty} \gamma^t R(s_t, \pi(s_t)) \, | s_0 = s \right].$$

The planning problem is centered on maximizing the expected return. In our setting, it exists a non necessarily unique policy $\pi^*$ such that $V^{\pi^*}(s) = \max_{\pi} V^{\pi}(s)$ simultaneoulsy for all states $s$. It is worth noting that the optimal value function $V^{\pi^*}$ (denoted as $V^*$) is the unique solution to the optimal Bellman Equation

$$V(s) = \max_{a \in \mathcal{A}} \left( R(s, a) + \gamma \sum_{s' \in \mathcal{S}} T(s, a, s') \cdot V(s') \right),$$
(1)

for all $s \in \mathcal{S}$ (Puterman 2014). Along this article, we denote by $(\mathcal{T}^* V)(s)$ the right term of Equation (1).

**Dynamic Programming**   Any value function can be computed recursively. Hence, for a given policy $\pi \in \mathcal{A}^{\mathcal{S}}$, we consider here the Bellman Operator

$$\mathcal{T}^{\pi} : V \in \mathbb{R}^{\mathcal{S}} \mapsto R^{\pi} + \gamma T^{\pi} \cdot V \in \mathbb{R}^{\mathcal{S}}.$$

with $R^{\pi}(s) = R(s, \pi(s))$ and $T^{\pi}(s, s') = T(s, \pi(s), s')$. This Bellman operator updates any value function $V$ relatively to the reward an transition functions. It contracts the function space and its iteration can lead to a value function solution of the Bellman equation $V = \mathcal{T}^{\pi} V$. One also considers the optimal Bellman Operator $\mathcal{T}^*$ defined by Equation (1).

So far, we have considered the state value function $V$, but a similar analysis can be conducted for the state-action value function $Q$ defined by

$$Q^{\pi}(s, a) = \mathop{\mathbb{E}}_{(s_t, a_t)_t} \left[ \sum_{t=0}^{\infty} \gamma^t R(s_t, \pi(s_t)) \, | s_0 = s, a_0 = a \right].$$

The optimal Bellman Operator in the $Q$-value case exists and is defined as

$$\mathcal{T}_Q^* : Q \in \mathbb{R}^{\mathcal{S} \times \mathcal{A}} \to R + \gamma T \cdot \max_{a \in \mathcal{A}}(Q) \in \mathbb{R}^{\mathcal{S} \times \mathcal{A}}.$$

The practical solving of a MDP, can be done either by maximizing the expected return $V^{\pi}$ for any state or by minimizing the Bellman residual namely $\|V - \mathcal{T}^* V\|_{\infty}$. The Dynamic Programming methods generally aim to decrease the Bellman residual. In Value Iteration algorithm (respectively $Q$-Value Iteration), one iterates the contracting optimal Bellman operator to approximate the fixed point solution of the optimal Bellman equation $V^* = \mathcal{T}^* V^*$ (respectively $Q^* = \mathcal{T}_Q^* Q^*$) (Puterman 2014). In Policy Iteration algorithm that alternate between finding the solution to $V = \mathcal{T}^{\pi} V$ and updating the current policy $\pi$ according to

$$\pi_{t+1}(s) \leftarrow \arg\max_{a \in \mathcal{A}} \left( R(s, a) + \gamma \sum_{s' \in \mathcal{S}} T(s, a, s') V^{\pi_t}(s') \right).$$

**State Abstraction**   The concept of constructing a new MDP through state aggregation has been explored in the literature, particularly in the examination of abstract MDPs (Li, Walsh, and Littman 2006). This involves considering a ground MDP, denoted as $\mathcal{M}_G = \langle \mathcal{S}, \mathcal{A}, T, R, \gamma \rangle$, and creating a new abstract MDP, denoted as $\mathcal{M}_A = \langle \mathcal{S}_A, \mathcal{A}, T_A, R_A, \gamma \rangle$, from it. We first define State Aggregation.

**Definition 1** (State aggregation). *Let $\mathcal{M}_G$ an MDP and $\mathcal{S} = \bigsqcup_{k=1}^{K} S_k$ a partition of the state space. Let assume we give a weight $\omega_k(s)$ to each state of a region $S_k$ relatively to the other states of this region $S_k$. The weights $\omega_k$ are positive and sum to 1 on each region. Moreover, if $s \notin S_k$, $\omega_k(s) = 0$. We finally store the weights into a matrix $\omega \in [0,1]^{K \times |\mathcal{S}|}$ and the state-region correspondence in a matrix $\phi$ such that $\phi[s,k] = \mathbb{1}_{s \in S_k}$. It is now possible to define a state aggregation by the tuple $((S_k)_k, \phi, \omega)$.*

When all states of a region are equally weighted, $\omega$ can be computed as follows: $\omega_k(s) = \frac{1}{|S_k|}$ which corresponds to $\omega = (\phi^T \cdot \phi)^{-1} \cdot \phi^T$ (Bertsekas, Castanon et al. 1988). Let us note that the following analysis can also be done in the general case of unequally weighted states. From now, the State Abstraction simply consists in building a new MDP from this aggregation.

**Definition 2** (Abstract MDP). *Let $\mathcal{M}_G$ an MDP and $((S_k)_k, \phi, \omega)$ a state aggregation. We represent each region $S_k$ by an abstract state $s_k$. The abstract MDP $\mathcal{M}_A$ can be therefore defined by $\mathcal{S}_A = \{s_k, 1 \leq k \leq K\}$, $\mathcal{A}_A = \mathcal{A}$, $T_A = \omega \cdot T \cdot \phi$ and $R_A = \omega \cdot R$.*

The interest of State Abstraction is therefore to reduce the size of the original MDP gathering state with similar properties like a close optimal value, a close optimal policy or a close optimal $Q$-value (Abel, Hershkowitz, and Littman 2016). It can be used to approximate the ground optimal policy but also to highlight a structure in the ground MDP.

**Approximate Dynamic Programming** While Dynamic Programming involves applying an operator to enhance the current solution, Approximate Dynamic Programming focuses on updating an approximated version of the value function (Powell 2007). In our context, we adopt the linear parameterization

$$V_\theta(s) = \sum_{k=1}^{K} \theta_k \mathbb{1}_{s \in S_k},$$

with $(S_k)_k$ a state aggregation. Those value function are constant over each region $S_k$. The approximate Bellman operator relative to this family of functions (denoted $\Pi\mathcal{T}^*$) is made of the optimal Bellman operator and a projection matrix $\Pi$ that averages the value on each region to obtain a piecewise constant value function.

**Definition 3** (Projected optimal Bellman operator (Tsitsiklis and Van Roy 1996)). *Let note $\mathcal{P}$ the set of value function that are piecewise constant relatively to $(S_k)_k$. Then, the operator $\Pi\mathcal{T}^*$ that checks*

$$\forall V \in \mathbb{R}^{\mathcal{S}}, \; \Pi\mathcal{T}^* \in \underset{\mathcal{T} \in \mathcal{P}}{\arg\min} \|\mathcal{T}V - \mathcal{T}^*V\|_2$$

*is the projected optimal Bellman Operator $\Pi\mathcal{T}^* = \phi \cdot \omega \cdot \mathcal{T}^*$ where $\phi$ and $\omega$ are described in Definition 1 of a state aggregation.*

In the following sections, we will consider each of the projected Bellman Operators $\Pi\mathcal{T}^*$, $\Pi\mathcal{T}_Q^*$ and $\Pi\mathcal{T}^\pi$ for any policy $\pi$ with $\Pi = \phi \cdot \omega$.

# Projected Bellman operators and State Abstraction

In what follows, we describe the relationship between the projected Bellman operators and State Abstraction. We first prove that a projected Bellman operator is exactly the Bellman operator of a smaller abstract MDP. As we want to implement those operators, we evaluate theirs complexity and compare it to the optimal Bellman operator $\mathcal{T}^*$.

**Projected Bellman Equations and abstract MDP** We are now interested in the unique solution of each of the equations $Q = \Pi\mathcal{T}_Q^*Q$ and $V = \Pi\mathcal{T}^\pi V$. We will namely prove that those projected equations are Bellman equations for the associated abstract MDP. Let us note that it does not generalize to the equation $V = \Pi\mathcal{T}^*V$. Indeed, as $Q$ and $V^\pi$ contains action information through $Q$ and $\pi$, any value function solution to $\tilde{V} = \Pi\mathcal{T}\tilde{V}$ is not necessarily associated with a piecewise constant policy. The solution of $V = \Pi\mathcal{T}^*V$ is therefore not necessarily the optimal value function of the abstract MDP.

It is now interesting to note that the functions solution to these equations $Q = \Pi\mathcal{T}_Q^*Q$ and $V = \Pi\mathcal{T}^\pi V$ are piecewise constant. Indeed, the operator $\Pi$ make the function being constant over the regions $(S_k)_k$. We are therefore adopting the following notations. Let $\tilde{V}$ a piecewise constant value function relatively to a partition $(S_k)_k$. The entries of $\tilde{V}$ are in a way redundant : for any state $s \in S_k$, $\tilde{V}(s)$ has the same value. We therefore build a contracted representation $\underline{V} \in \mathbb{R}^K$ which contains a single value for each region: $\forall s \in S_k, \tilde{V}(s) = \underline{V}(k)$. This new value function $\underline{V}$ can be a value function to the associated abstract MDP. Moreover, it is possible to switch between $\tilde{V}$ and $\underline{V}$ using the relations $\tilde{V} = \phi \cdot \underline{V}$ and $\underline{V} = \omega \cdot \tilde{V}$. We use similar notation concerning the $Q$-value using $\tilde{Q}$ and $\underline{Q}$.

In the following proposition, we suggest that the solution to $Q = \Pi\mathcal{T}_Q^*Q$ is also the optimal $Q$-value function of the abstract MDP.

**Proposition 1.** *Let $((S_k)_k, \phi, \omega)$ be an aggregation of the state space. Let $\tilde{Q} = \phi \cdot \underline{Q}$ be the unique solution to the $Q$-projected optimal Bellman equation $Q = \Pi\mathcal{T}_Q^*Q$. Then $\underline{Q}$ is the optimal $Q$-value function of the abstract MDP $\mathcal{M}_A$ described in Definition 2.*

The proof simply consists in establishing that the equation $Q = \Pi\mathcal{T}_Q^*Q$ can be written as the optimal Bellman equation $Q = \mathcal{T}_Q^*Q$ for the abstract MDP.

*Proof.* Let $\tilde{Q} = \phi \cdot \underline{Q}$ the unique solution to $Q = \Pi\mathcal{T}_Q^*Q$. Let $Q_A^*$ the optimal $Q$-value of the abstract MDP $\mathcal{M}_A$. Let show that those $Q$-value are the solution to the same equation. The

equation $\tilde{Q} = \Pi\mathcal{T}_Q^*\tilde{Q}$ can be written :

$$\tilde{Q} = \Pi\mathcal{T}_Q^*\tilde{Q}$$
$$\iff \phi \cdot \mathbf{Q} = \phi \cdot \omega \cdot \mathcal{T}_Q^*\left(\phi \cdot \mathbf{Q}\right)$$
$$\iff \mathbf{Q} = \omega \cdot \mathcal{T}_Q^*\left(\phi \cdot \mathbf{Q}\right)$$
$$\iff \mathbf{Q} = \omega \cdot R + \gamma\omega \cdot T \cdot \phi \cdot \max_{a\in\mathcal{A}}\left(\mathbf{Q}\right)$$
$$\iff \mathbf{Q} = R_A + \gamma T_A \cdot \max_{a\in\mathcal{A}}\left(\mathbf{Q}\right) .$$

which is precisely the optimal Bellman equation for the abstract MDP $\mathcal{M}_A$. As the solution to each of the equation is unique, we can conclude that $\mathbf{Q} = Q_A^*$. $\qquad\square$

As in (Abel, Hershkowitz, and Littman 2016), we focus here on $\pi_A$ an arbitrary policy on the abstract state space $\mathcal{S}_A$. We define its generalization $\pi_G$ to the ground state space $\mathcal{S}$, by the piecewise constant policy given by:

$$\pi(s) = \pi_A(s_k)\,, \ \forall s \in S_k\,, \quad \forall k \in [\![1\ ;\ K]\!].$$

Proposition 1 has an equivalent for the $\mathcal{T}^\pi$ operator. Hence in Proposition 2, we state that the value of any abstract policy $\tilde{V}^{\pi_A}$ is the solution of a projected Bellman equation $\tilde{V}^{\pi_A} = \Pi\mathcal{T}^{\pi_A}\tilde{V}^{\pi_A}$ at the ground level.

**Proposition 2.** *Let $((S_k)_k, \phi, \omega)$ an aggregation of the state space. Let $\pi_A : \mathcal{S}_A \mapsto \mathcal{A}$ an arbitrary policy and $\pi_G : \mathcal{S} \mapsto \mathcal{A}$ its generalization to the ground state space $\mathcal{S}$. Then, the value of this policy on the abstract MDP $\underline{V}^{\pi_A}$ is the solution of the following projected Bellman equation*

$$\phi \cdot \underline{V}^{\pi_A} = \Pi\mathcal{T}^{\pi_G}\left(\phi \cdot \underline{V}^{\pi_A}\right).$$

The proof still relies on the unicity of the solution of a fixed-point equality.

*Proof.* In the following, we prove the equivalence of the equations

$$\tilde{V} = \Pi\mathcal{T}^{\pi_G}\tilde{V} \qquad \text{and} \qquad \underline{V} = \mathcal{T}^{\pi_A}\underline{V}$$

which suffices to conclude on the proposition.

$$\tilde{V} = \Pi\mathcal{T}^{\pi_G}\tilde{V} \iff \phi \cdot \underline{V} = \Pi\mathcal{T}^{\pi_G}\left(\phi \cdot \underline{V}\right)$$
$$\iff \phi \cdot \underline{V} = \phi \cdot \omega \cdot \left(R^{\pi_G} + \gamma \cdot T^{\pi_G} \cdot \phi \cdot \underline{V}\right)$$
$$\iff \underline{V} = \omega \cdot R^{\pi_G} + \gamma \cdot \omega \cdot T^{\pi_G} \cdot \phi \cdot \underline{V}$$
$$\iff \underline{V} = R_A^{\pi_A} + \gamma \cdot T_A^{\pi_A}\underline{V} \iff \underline{V} = \mathcal{T}^{\pi_A}\underline{V}$$

Those equivalences imply the wanted equality and therefore on the property. $\qquad\square$

As we proved here that the solution of projected Bellman equation if the optimal value function of an abstract MDP, we now study the complexity of iterating a projected Bellman Operator.

**Iterations of projected Bellman Operators** In this part, we prove the convergence of any sequence of value functions (or $Q$ value function) on which we iterate any projected Bellman Operator.

**Proposition 3.** *1. Let $Q_0 \in \mathbb{R}^{\mathcal{S}\times\mathcal{A}}$ be an arbitrary $Q$-value function and let the iteration $Q_{t+1} \leftarrow \Pi\mathcal{T}_Q^*Q_t$. Then the series $(Q_t)_{t\in\mathbb{N}}$ converges to the unique solution to the projected optimal Bellman equation $Q = \Pi\mathcal{T}_Q^*Q$.*

*2. Let $\pi \in \mathcal{A}^\mathcal{S}$ be an arbitrary policy. Let $V_0 \in \mathbb{R}^\mathcal{S}$ any value function, and let the iteration $V_{t+1} \leftarrow \Pi\mathcal{T}^\pi V_t$. Then $(V_t)$ converges to the unique solution to the projected Bellman equation $V = \Pi\mathcal{T}^\pi V$.*

In (Bertsekas 2018) was established the contraction property of the operator $\Pi\mathcal{T}^*$. We generalize it to $\Pi\mathcal{T}_Q^*$ and $\Pi\mathcal{T}^\pi$ for any policy $\pi$ to prove Proposition 3.

**Proposition 4.** *The operators $\Pi\mathcal{T}_Q^*$ and $\Pi\mathcal{T}^\pi$ for any policy $\pi$ are contracting.*

*Proof.* The proof relies on the contraction induced by the Bellman operator and on the following inequality true for any $\mathcal{T} \in \{\mathcal{T}_Q^*, \mathcal{T}^\pi\}$:

$$\|\phi \cdot \omega \cdot (\mathcal{T}V - \mathcal{T}V')\|_\infty \leq \|\mathcal{T}V - \mathcal{T}V'\|_\infty$$

as $\phi$ simply repeat the entries in any vector $V \in \mathbb{R}^\mathcal{S}$. This property can be applied to $\Pi\mathcal{T}_Q^*$ and $\Pi\mathcal{T}^\pi$ for any policy $\pi$ to conclude the proof. $\qquad\square$

From now on, iterating any of the operator $\mathcal{T}^*$, $\Pi\mathcal{T}^*$, $\Pi\mathcal{T}_Q^*$ or $\Pi\mathcal{T}^\pi$ makes any value function converge to a unique piecewise constant final value function. In the following, we will be interested in the complexity of the computation of the solution of the projected Bellman equation and will propose a bound on the error to the optimal value function depending on the specific aggregation and on the current value.

**Projected Operators complexity** Now, we will consider the complexity of computing the projected Bellman operators $\Pi\mathcal{T}^*$, $\Pi\mathcal{T}_Q^*$ and $\Pi\mathcal{T}^{\pi_G}$ for any piecewise constant policy $\pi_G$.

**Proposition 5.** *The complexity of the computation of the projected Bellman operators $\Pi\mathcal{T}_Q^*$ and $\Pi\mathcal{T}^{\pi_G}$ for any piecewise constant policy $\pi_G$ are respectively $O(K^3\,|\mathcal{A}|)$ and $O(K^3)$.*

*Proof.* As $\Pi\mathcal{T}_Q^*$ and $\Pi\mathcal{T}^{\pi_G}$ can be viewed as the Bellman operators $\mathcal{T}_{\mathbf{Q}}^*$ and $\mathcal{T}^{\pi_A}$, then theirs complexity can be computed from the abstract MDP point of view. We therefore deduce their complexity from the matrix computations

$$\mathcal{T}_{\mathbf{Q}}^*\left(\mathbf{Q}\right) = R_A + \gamma \cdot T_A \cdot \max_{a\in\mathcal{A}}\left(\mathbf{Q}\right)$$

and

$$\mathcal{T}^{\pi_A}\left(\underline{V}\right) = R_A^{\pi_A} + \gamma \cdot T_A^{\pi_A} \cdot \underline{V}$$

knowing that the complexity of the product $M \cdot N$, with $M \in \mathbb{R}^{l\times m}$, and $N \in \mathbb{R}^{m\times n}$ is equal to $l \cdot m \cdot n$. $\qquad\square$

From now, we consider the computation complexity of the projected optimal Bellman operator $\Pi\mathcal{T}^*$ assuming we will iterate it.

**Proposition 6.** *For any piecewise constant value function $\tilde{V}$, the number of operations to compute $\Pi\mathcal{T}^*\tilde{V}$ is $O(|\mathcal{S}|^2\,K\,|\mathcal{A}|)$.*

*Proof.* Considering

$$\Pi\mathcal{T}^*\left(\phi \cdot \underline{\mathbf{V}}\right) = \phi \cdot \omega \max_{a \in \mathcal{A}}\left(R + \gamma \cdot T \cdot \phi \cdot \underline{\mathbf{V}}\right)$$

the precomputation of the matrix product $T \cdot \phi \in \mathbb{R}^{|\mathcal{S}| \times \mathcal{A} \times K}$ allows the matrix product $(T \cdot \phi) \cdot \underline{\mathbf{V}}$ to have a complexity of $O(|\mathcal{S}|^2 K |\mathcal{A}|)$. $\qquad\square$

The complexities of $O(|\mathcal{S}|^2 K |\mathcal{A}|)$ for $\Pi\mathcal{T}^*$, $O(K^3 |\mathcal{A}|)$ for $\Pi\mathcal{T}_Q^*$, and $O(K^3)$ for $\Pi\mathcal{T}^\pi$ are noteworthy when contrasted with the $O(|\mathcal{S}|^3 |\mathcal{A}|)$ complexity for $\mathcal{T}^*$. Having established that computing projected operators is more straightforward than traditional ones, we introduce an algorithm to systematically disaggregate regions into smaller ones, facilitating the evolution of a piecewise constant value function.

## Progressive Disaggregation process

In this section, we first establish a bound between a given piecewise constant value function and the optimal value function of any MDP. This bound depends on the aggregation quality (*i.e.* the capacity to aggregate states with the same value function) and the projected Bellman residual $\tilde{V} - \Pi\mathcal{T}\tilde{V}$ but does not use the optimal value function. We then provide the Progressive Disaggregation algorithm which is based on this bound : we iteratively improve the aggregation quality (reducing one term of the bound of theorem 1) and decrease the projected Bellman residual by applying the projected Bellman operator.

**Theorem 1** (optimal Error Bound with arbitrary partition)**.** *Let any piecewise constant value function $\tilde{V} \in \mathbb{R}^{\mathcal{S}}$. Its distance to the optimal value function $V^*$ can be bounded as follows:*

$$\begin{aligned}\|\tilde{V} - V^*\|_\infty \leq &\frac{1}{1-\gamma} \max_{1 \leq k \leq K} \mathrm{Span}_{S_k}\left(\mathcal{T}^*\tilde{V}\right) \\ &+ \frac{1}{1-\gamma}\|\tilde{V} - \Pi\mathcal{T}^*\tilde{V}\|_\infty,\end{aligned} \quad (2)$$

*where* $\mathrm{Span}_{S_k}(V) := \max_{s \in S_k} V(s) - \min_{s \in S_k} V(s)$.

*Proof.* We mainly use the classical inequality:

$$\forall V \in \mathbb{R}^{\mathcal{S}}, \; \|V - V^*\|_\infty \leq \frac{1}{1-\gamma}\|V - \mathcal{T}^*V\|_\infty$$

and also the following one:

$$\forall V \in \mathbb{R}^{\mathcal{S}}, \; \|V - \Pi V\|_\infty \leq \max_{1 \leq k \leq K} \mathrm{Span}_{S_k}(V).$$

Concatenating inequalities, we get:

$$\begin{aligned}\|V^* - \tilde{V}\|_\infty &\leq \frac{1}{1-\gamma}\|\tilde{V} - \mathcal{T}^*\tilde{V}\|_\infty \\ &\leq \frac{1}{1-\gamma}\left(\|\Pi\mathcal{T}^*\tilde{V} - \mathcal{T}^*\tilde{V}\|_\infty + \|\tilde{V} - \Pi\mathcal{T}^*\tilde{V}\|_\infty\right) \\ &\leq \frac{1}{1-\gamma}\left(\max_{1 \leq k \leq K} \mathrm{Span}_{S_k}\left(\mathcal{T}^*\tilde{V}\right) + \|\tilde{V} - \Pi\mathcal{T}^*\tilde{V}\|_\infty\right)\end{aligned}$$

$\qquad\square$

We furthermore note that $\max_{1 \leq k \leq K} \mathrm{Span}_{S_k}\left(\mathcal{T}^*\tilde{V}\right)$ measures how much the aggregation groups states having the same value and that $\|\tilde{V} - \Pi\mathcal{T}^*\tilde{V}\|_\infty$ estimates the optimality of the current piecewise value function relatively to the projected Bellman operator.

**Corollary 1.** *Inequality 2 can also be formulated using $\Pi\mathcal{T}_Q^*$ and $\Pi\mathcal{T}^{\pi_G}$ for any piecewise constant policy $\pi_G$.*

We therefore propose an algorithm with initialization $\tilde{V}_0 = (0)_{s \in \mathcal{S}}$ and a unique region $S_0 = \{\mathcal{S}\}$. We then iterate the two following steps successively:

- Apply $\Pi\mathcal{T}^*$ until $\|\tilde{V} - \Pi\mathcal{T}^*\tilde{V}\|_\infty$ is smaller than $\epsilon$
- Compute $\mathcal{T}^*V_t$. Divide each region until $\max_{s \in S_k} V_{t+1} - \min_{s \in S_k} V_{t+1}$ is smaller than $\epsilon$ for each region $k \in [\![1 \; ; \; K]\!]$.

When applying this process, we separate states having different trajectories through value iteration. Moreover note that the $\Pi\mathcal{T}^*$ operator changes at each region division step. The goal is also to take advantage of the time savings from the projected Bellman operator application compared to the optimal ground one.

## Proposed algorithms

In this section, we provide the pseudocode for the algorithm that we described previously. We precise its adaptation to $Q$-value iteration and Modified Policy Iteration. We then prove the convergence of the algorithm and lead a complexity analysis to conclude on its performance condition.

**Formulation** In Algorithm 1, we describe the *Progressive Disaggregation Value Iteration* (**PDVI**) process. It can be summarized into *while the bound of the Theorem 1 is not lower than $\epsilon$, then alternate between dividing heterogeneous regions and updating the piecewise constant value function $\tilde{V}$.* We managed to make the region division to have a $O(|\mathcal{S}|)$ complexity. This allows to profit from piecewise constant update savings.

---

**Algorithm 1:** Progressive Disaggregation Value Iteration

**Input**: $\mathcal{M} = \langle \mathcal{S}, \mathcal{A}, T, R, \gamma \rangle, \epsilon > 0$
**Output**: A value $V$, an aggregation $(S_k)_k$ of the state space
1: $K := 1, S_1 := \mathcal{S}, \underline{\mathbf{V}}_0 := (0)_{1 \leq k \leq K}$
2: **while**

$$\|\underline{\mathbf{V}}_t - \Pi\mathcal{T}^*\underline{\mathbf{V}}_t\|_\infty + \max_{1 \leq k \leq K} \mathrm{Span}_{S_k}(\mathcal{T}^*\underline{\mathbf{V}}_t) > 2\epsilon$$

   **do**
3: $\quad V_{t+1} := \mathcal{T}^*(\phi \cdot \underline{\mathbf{V}}_t)$
4: $\quad$**if** $\max_{1 \leq k \leq K} \mathrm{Span}_{S_k}(V_{t+1}) > \epsilon$ **then**
5: $\quad\quad (S_k)_k = \mathrm{UpdateRegion}(k, V_k, (S_k)_k, \epsilon)$
6: $\quad$**end if**
7: $\quad$**while** $\|\underline{\mathbf{V}}_t - \Pi\mathcal{T}^*\underline{\mathbf{V}}_t\|_\infty > \epsilon$ **do**
8: $\quad\quad \underline{\mathbf{V}}_{t+1} \leftarrow \Pi\mathcal{T}^*\underline{\mathbf{V}}_t$
9: $\quad$**end while**
10: **end while**
11: **return** $(\underline{\mathbf{V}}, (S_k)_k)$

---

Algorithm 2: UpdateRegions

**Input**: $V, (S_k)_k, \epsilon$
**Output**: Updated partition $\{S'_k\}$

1: **for** $l \in [\![1 \; ; \; K]\!]$ **do**
2:  **if** $\max_{s \in S_l} V - \min_{s \in S_l} V > \epsilon$ **then**
3:   $(S_k)_k = (S_k)_k \backslash S_l$
4:   **for** $p \in [\![0 \; ; \; \lceil \frac{1}{\epsilon}(\max V_{|S_l} - \min V_{|S_l})\rceil]\!]$ **do**
5:    $I_p := [\min V_{|S_l} + p.\epsilon, \min V_{|S_l} + (p+1).\epsilon]$
6:    $S_p := \{s \in S_l, V(s) \in I_p\}$
7:    **if** $S_p \neq \emptyset$ **then**
8:     $(S_k)_k := (S_k)_k \sqcup S$
9:    **end if**
10:   **end for**
11:  **end if**
12: **end for**
13: **return** Updated partition $\{S_k\}$

As this algorithm consists in iterating $\Pi\mathcal{T}^*$ and dividing regions along $\mathcal{T}^*V$, we generalize it to the $Q$-value process by applying $\Pi\mathcal{T}_Q^*$ and divide the regions along $\mathcal{T}_Q^*Q$. We name this new algorithm *Progressive Disaggregation Q-Value Iteration* (**PDQVI**). In the region division step, we ensure
$$\max_{s \in S_k, a \in \mathcal{A}} Q - \min_{s \in S_k, a \in \mathcal{A}} Q \leq \epsilon.$$
Moreover, PDQVI provide a state abstraction gathering states with close optimal $Q$-value $Q^*$.

We also propose a Modified Policy Iteration-type algorithm named *Progressive Disaggregation Policy Iteration* (**PDPI**). In Modified Policy Iteration, we start from an arbitrary given policy. We then iteratively evaluate its value function $V^\pi$ (solution of $V^\pi = \mathcal{T}^\pi V^\pi$) and update the policy using $V^\pi$. In PDPI, we changed the Policy Evaluation part into a disaggregation process, progressively dividing regions and evaluating the solution of $V = \Pi\mathcal{T}^\pi V$ at the same time.

**Convergence property** In this part, we state a guarantee of convergence for PDVI, PDQVI and PDPI and characterize the aggregation provided by the algorithms.

**Proposition 7** (Convergence guarantee for PDVI). *Algorithm 1 finishes in a finite number of steps. Let $(\underline{V}, (S_k)_k)$ be the value and the abstraction returned. Then :*
$$\|\phi \cdot \underline{V} - V^*\|_\infty \leq \frac{2\epsilon}{1 - \gamma}$$
*where the precision $\epsilon$ is an input of the algorithm. Moreover, for all $k \in [\![1 \; ; \; K]\!]$, for all $s, s' \in S_k$, we have*
$$|V^*(s) - V^*(s')| \leq \frac{4\epsilon}{1 - \gamma}.$$

We stated here that PDVI converges with an accuracy similar to Value Iteration and aggregates states having close optimal value.

*Proof.* At first, let prove that Algorithm 1 stops within $|\mathcal{S}| + 1$ steps by contradiction. Let assume that
$$\|\underline{V}_t - \Pi\mathcal{T}^*\underline{V}_t\|_\infty + \max_{1 \leq k \leq K} \text{Span}_{S_k}(\mathcal{T}^*\underline{V}_t) > 2\epsilon$$

after $|\mathcal{S}| + 2$ steps. As at the end of each step $t \in [\![0 \; ; \; |\mathcal{S}| + 2]\!]$, $\|\underline{V}_t - \Pi\mathcal{T}^*\underline{V}_t\|_\infty \leq \epsilon$ (due to the lines 7-9 condition), then it follows that $\max_{1 \leq k \leq K} \text{Span}_{S_k}(\mathcal{T}^*\underline{V}_t) > \epsilon$ as the *while* condition is not fulfilled. We therefore deduce that for each of the steps $t \in [\![1 \; ; \; |\mathcal{S}| + 1]\!]$, a disaggregation step has occured. Given that for each disaggregation step, the number of region strictly increases, we deduce that at step $t = |\mathcal{S}| + 1$, the state aggregation is made of $|\mathcal{S}| + 1$ regions. This can not occur. So, the final precision condition
$$\|\phi \cdot \underline{V} - V^*\|_\infty \leq \frac{2\epsilon}{1 - \gamma}$$
is then ensured by the while loop condition combined with Theorem 1.

Finally, let us show that the regions $(S_k)_k$ group states having close optimal value. Let $k \in [\![1 \; ; \; K]\!]$ any region and $s, s' \in S_k$. We observe that
$$\|V^*(s) - V^*(s')\|_\infty$$
$$\leq \|V^*(s) - \tilde{V}(s)\|_\infty + \|\tilde{V}(s) - V^*(s')\|_\infty$$
$$= \|V^*(s) - \tilde{V}(s)\|_\infty + \|\tilde{V}(s') - V^*(s')\|_\infty$$
$$\leq \frac{4\epsilon}{1 - \gamma}.$$
The equality comes from $\tilde{V}(s) = \tilde{V}(s')$ as $s$ and $s'$ are in the same region. The last inequality can be stated using the final precision of the algorithm. $\square$

We also mention that PDQVI aggregates states having close optimal $Q$-value $Q^*$ and PDPI also groups states having same optimal value $V^*$. Both converge and provide optimal $Q$-value and optimal policies following the same steps for the proof. We add that the proof of the policy-based disaggregation algorithm convergence contains some subtilty, especially when it is necessary to keep the value function $V^\pi$ from one policy evaluation to the next one. We finally state that PDQVI and PDPI still converge in the expected total-reward criterion $\gamma = 1$. This convergence result can only be checked with the assumption $R \geq 0$ as we use the convergence properties of Value Iteration and Policy Iteration in the expected total-reward case (Puterman 2014).

## Complexity Analysis

We provide here a complexity analysis for PDVI, PDQVI and PDPI and see that the worst-case complexity of those algorithms can be higher than the traditional ones. This characterization will also explain why our disaggregation method can be outperformed by traditional ones on some specific models that we identify. Hence, to prove the bounds that we exhibit, one consider an instance of chained states inspired from the Four Room model described in (Hengst 2012).

**Proposition 8** (Disaggregation algorithm complexity). *Let assume that any Value Iteration-like algorithm takes $n$ steps to reach an $\epsilon$-close optimal value. To reach an $\epsilon$-optimal value function, our algorithms require the following number of operations.*

| Algorithm | PDVI | PDQVI | PDPI |
|---|---|---|---|
| Operations | $O(n\,|\mathcal{S}|^4\,|\mathcal{A}|)$ | $O(n\,|\mathcal{S}|^4\,|\mathcal{A}|)$ | $O(n\,|\mathcal{S}|^4)$ |

*Proof.* To evaluate the complexity of PDVI,

1. We assume that we perform $|\mathcal{S}|$ disaggregation steps and count operations made in this case,
2. We exhibit a MDP where PDVI effectively performs $|\mathcal{S}|$ disaggregation steps.

Let consider the execution of $|\mathcal{S}|$ iterations of the $\Pi\mathcal{T}^*$ operator. According to the Proposition 6, each iteration of $\Pi\mathcal{T}^*$ takes $O(K\,|\mathcal{S}|^2\,|\mathcal{A}|)$ operations. We assume it takes $n$ operations to approximate the solution of the projected optimal Bellman equation $\tilde{V} = \Pi\mathcal{T}^*\tilde{V}$ with an accuracy $\epsilon$. The total number of operations through the $|\mathcal{S}|$ iterations can be estimated as $O(|\mathcal{S}|^2\,|\mathcal{A}|\sum_{K=1}^{|\mathcal{S}|} K) = O(|\mathcal{S}|^4\,|\mathcal{A}|)$.

From now, let us exhibit the case where we indeed perform $|\mathcal{S}|$ aggregation steps. Let consider $|\mathcal{S}|$ states linked like a chain with two actions (left and right) for each, like in the Figure 1. The leftmost state $s_1$, the left action keeps us at $s_1$ and idem for the rightmost state $s_{|\mathcal{S}|}$ and right action. The state $s_1$ is considered as an exit state and is therefore absorbing. Concerning the reward, each action made in a state different from the exit generates a $-1$ reward.

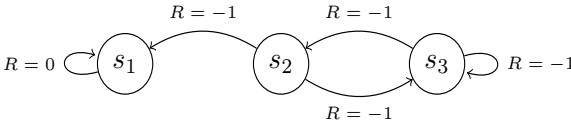

Figure 1: Worst case for Progressive Disaggregation Value Iteration with 3 states.

Let us consider the execution of $PDVI$ on this MDP. The optimal value function is

$$V^* = \begin{pmatrix} 0 & -1 & \dots & -\sum_{s=0}^{|\mathcal{S}|}\gamma^s \end{pmatrix}$$

The different regions considered through PDVI are $\{s_0, \dots, s_{|\mathcal{S}|}\}$, thus $\{s_0\}\sqcup\{s_1\cdots, s_{|\mathcal{S}|}\}$ until $\{s_0\}\sqcup\{s_1\}\sqcup\dots\sqcup\{s_{|\mathcal{S}|}\}$ : starting from the exit the partition discover each state iteratively. We finally proved the complexity estimation considering this specific instance.

The same kind of argument are necessary to evaluate the complexity of Progressive Disaggregation $Q$-Value Iteration and Progressive Disaggregation Policy Iteration. $\qquad\square$

Let note that Value Iteration algorithm takes at most $O(n\,|\mathcal{S}|^3\,|\mathcal{A}|)$ operations (Feinberg and He 2020) and Policy Iteration requires $O(|\mathcal{S}|^3)$ (Littman, Dean, and Kaelbling 1995). We obviously lose in complexity during disaggregation in some specific cases that we detailed here. Nevertheless we claim (based on numerical experiments performed later) that these worst-case bounds are not reached in practice and that the average complexity of our algorithm is much better.

## Numerical Results

We conducted a benchmark of our solving methods on three scalable models. To run this benchmark, we implemented several solvers. We compared PDVI, PDQVI and PDPI to usual Value Iteration, Modified Policy Iteration, as well as Bertsekas Adaptive Aggregation (Bertsekas, Castanon et al. 1988) and Chen Adaptive Aggregation (Chen et al. 2022). Our comparison with a diverse set of solving methods show that our disaggregation algorithms outperforms other methods on most of the models.

We selected configurable models with variable state space and action space sizes. We evaluated MDPs on a randomly generated transition matrix, a toy model (four rooms), and a real-life problem. The Random MDPs are commonly used in the literature to benchmark solvers (Archibald, McKinnon, and Thomas 1995; Bhatnagar et al. 2009). The Four Rooms model is a stochastic shortest path model (Hengst 2012; Sutton and Barto 2018) that we scaled to explore the state space size impact. Finally, we faced a real-life queue management situation with scalable servers and queue sizes (Ohno and Ichiki 1987; Tournaire et al. 2022) already used for benchmark in (Puterman 2014).

We ran the solving process on one thread of a CPU Intel Xeon Gold 6154 @ 3.00GHz. The code has been written in Python using `numpy` and `scipy` libraries to encode sparse matrices and the experiment used at most 16GB of RAM. Chen Adaptive Aggregation method was far behind the other value-based method up to a factor 2, we therefore decided to only keep other methods in the following results.

**Random models** Our slicing strategy gave its best on random models. We drew random distributions $T(s, a, .)$ on $\mathcal{S}$ for any $(s, a) \in \mathcal{S} \times \mathcal{A}$. We also build $R$ with random coefficients in $[0, 1]$. We set $|\mathcal{S}| = 500$ and $|\mathcal{A}| = 50$ and a variable proportion of nonzero entries (named density) in the transition matrix. As the density of the transition matrix impacted the most the optimal value function shape, we set a maximum of diversity in this parameter going from $1\%$ (almost empty matrix) to $65\%$ (two over three pairs of state are connected by a nonzero transition) of nonzero entries.

As shown in Table 1, Progressive Disaggregation methods demonstrate their advantages for both value and policy-based approaches, aligning with the theoretical analysis in the previous section. Indeed, small densities of $T$ induce independent states while higher densities of $T$ smooth the optimal value functions.

**Real model** We considered a Tandem Queue situation inspired from a real-world server operation (Ohno and Ichiki 1987). Here, the agent manages to scale several servers relatively to the load of a system made of two tandem queues with parallel servers. To this end, it is possible to activate or deactivate servers. There are 3 actions (add, keep or remove a server) for each queue which gives 9 actions in total. We could scale here the size of the queue and the size of the server to adjust the state space dimension. We present the results for $|\mathcal{S}| \in \{8100, 12544\}$. According to common hypothesis in this domain, we chose a queue size (15 and 16) greater than the server size (6 and 7) (Tournaire et al. 2022).

| Density | VI | PDVI | PDQVI | MPI | PDPI | Bertsekas |
|---|---|---|---|---|---|---|
| 1% | $113.3 \pm 1.0$ | $6.6 \pm 0.5$ | $8.0 \pm 0.4$ | $3.0 \pm 1.25$ | $1.09 \pm 0.23$ | $2.8 \pm 0.6$ |
| 10% | $300.3 \pm 10.9$ | $7.5 \pm 0.1$ | $15.2 \pm 0.3$ | $1.65 \pm 0.46$ | $1.57 \pm 0.45$ | $2.5 \pm 0.34$ |
| 25% | $751.7 \pm 16.0$ | $6.2 \pm 0.6$ | $24.1 \pm 0.8$ | $1.17 \pm 0.08$ | $0.72 \pm 0.11$ | $1.5 \pm 0.4$ |
| 45% | $1397.7 \pm 23.7$ | $7.6 \pm 1.3$ | $36.3 \pm 1.7$ | $1.83 \pm 0.32$ | $0.61 \pm 0.21$ | $2.0 \pm 0.2$ |
| 65% | $1915.4 \pm 54.2$ | $6.7 \pm 0.4$ | $50.3 \pm 3.6$ | $2.86 \pm 1.03$ | $1.57 \pm 0.74$ | $3.3 \pm 0.7$ |

Table 1: Time (in seconds) for solving Random MDPs with variable transition matrix density. $\mathcal{S} = 500$, $\mathcal{A} = 50$, $\gamma = 0.99$, final precision $10^{-2}$, average runtime on 10 experiments.

| $|\mathcal{S}|$ | VI | PDVI | PDQVI | MPI | PDPI | Bertsekas |
|---|---|---|---|---|---|---|
| 8100 | $12.1 \pm 0.5$ | $8.0 \pm 1.3$ | $15.3 \pm 0.7$ | $1442.5 \pm 39.2$ | $267.5 \pm 5.6$ | $1626.1 \pm 13.4$ |
| 12544 | $41.5 \pm 0.8$ | $18.8 \pm 1.8$ | $35.3 \pm 1.6$ | $4211.0 \pm 63.1$ | $994.7 \pm 6.3$ | $3577.2 \pm 14.8$ |

Table 2: Time (in seconds) to solve Discrete Tandem Queue models with variable state space size. ($|\mathcal{S}| \in \{8100, 12544\}$, $|\mathcal{A}| = 3$, $\gamma = 0.99$, final precision $10^{-2}$, average run time on 10 experiments.)

| $|\mathcal{S}|$ | VI | PDVI | PDQVI | MPI | PDPI | Bertsekas |
|---|---|---|---|---|---|---|
| 36 | $2.72 \pm 0.0$ | $7.46 \pm 0.4$ | $103.28 \pm 0.7$ | $2 \pm 1$ | $1 \pm 0.1$ | $1 \pm 0.5$ |
| 100 | $3.63 \pm 0.1$ | $6.77 \pm 1.7$ | $267.63 \pm 2.6$ | $18 \pm 3$ | $2 \pm 0.7$ | $19 \pm 0.9$ |
| 196 | $3.57 \pm 0.4$ | $9.25 \pm 2.7$ | $276.04 \pm 2.5$ | $29 \pm 4$ | $3 \pm 0.4$ | $29 \pm 0.9$ |
| 324 | $10.25 \pm 0.8$ | $14.16 \pm 5.0$ | $456.31 \pm 7.9$ | $47 \pm 7$ | $10 \pm 1.2$ | $47 \pm 0.6$ |

Table 3: Time (in seconds) for solving Four Room models with variable state space size. $|\mathcal{S}| \in \{36, 100, 196, 324\}$, $|\mathcal{A}| = 4$, $\gamma = 0.999$, final precision $10^{-3}$, average run time on 10 experiments.

In the results shown in Table 2, the disaggregation method still outperforms Modified Policy Iteration, Value Iteration and Bertsekas version of Modified Policy Iteration. This model is particularly fastly solved by Value Iteration method and the partition update accelerate the process in the first steps. Indeed, the individual update of Value Iteration is transformed into a region update of value in Progressive Disaggregation Value Iteration.

**Classical model** We finally considered the well-known grid model four rooms (Hengst 2012). This model is made of four rooms set out in square ($5 \times 5$ states for each room) with doors connecting pairs of rooms. As a stochastic shortest path model, the goal is to reach the exit (a given state of the grid) and each action in $(N, S, E, W)$ leads to the next close state with a probability $.8$ (if there is no wall in this direction), otherwise we stay in place. To adapt it into a discounted model, the agent restarts at the beginning when reaching the exit. We scaled the model to greater room sizes in order to make it more complex. The model is therefore very sparse: $2.|\mathcal{S}|.|\mathcal{A}|$ coefficient are non-zero in the transition matrix of size $|\mathcal{S}|^2.|\mathcal{A}|$ which implies a sparsity of at least $98\%$ for our instance. The slicing algorithm performs better and better as the state space dimension increases for the policy-based version.

In this experiment, we increased the discount factor up to 0.999 and the final precision to $10^{-3}$ so that the value-based methods struggle a bit more and are outperformed by policy-based ones. The results are shown in Table 3. Note that, the partition found by our method gathers states which are equidistant from the exit.

## Conclusion

Solving and approaching the exact MDP solution remain questions that deeply depend on the problem structure. In this context, we present an approximation method that combines State Abstraction and aggregation methods to accelerate traditional dynamic programming algorithms.

We focused on three main aspects. Initially, we established a robust connection between the projected Bellman operator and the abstract MDP's Bellman operator, extending it to the $Q$-value case and introducing a policy-based version. Following that, we presented a bound on the distance to the optimal value function based on a given state abstraction, leading to a progressive disaggregation process to refine state partitions. Our algorithm tests demonstrated the effectiveness of this approach, particularly in solving MDPs with dense transition matrices. Compared to Modified Policy Iteration and other Adaptive Aggregation methods, the policy-based approach significantly outperformed in solving realistic MDP instances with well-known models.

Our approach could benefit from further testing on MDP instances. Additionally, algorithmic enhancements could be introduced to switch to traditional dynamic programming methods when the partition proves inefficient. As disaggregation methods faces challenges primarily in cases close to the shortest path problem, it is crucial to tailor our approach to be more specific to this type of problem. Taking inspiration from the approximation of Tsitsiklis and Castañon, this method could also be combined with progressive disaggregation process. Future work should also investigate generalizations to the model-free Reinforcement Learning problem, incorporating not only state grouping but also the approximation of the state space using Deep Learning methods.

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
