# OpenReview forum: "Progressive State Space Disaggregation for Infinite Horizon Dynamic Programming"
_icaps-conference.org/ICAPS/2024/Conference — ICAPS 2024_

### Official Review · Reviewer_4srN · 2024-01-19

**Significance And Importance:** 2
**Soundness:** 4
**Novelty:** 2
**Clarity:** 3
**Overall Evaluation:** 1
**Confidence:** 4

**Weaknesses:**

1: Minor weaknesses that are easily fixable.

**Contributions Of The Paper:**

The paper studies disaggregation mechanisms for infinite-horizon dynamic programs. More precisely, the authors formulate a state-abstraction MDP that is derived by merging selected states into regions, and investigate its optimality structure as a projected Bellman equation. Based on this theory, the paper establishes the complexity of computing an optimal piecewise constant policy as a function of the number of regions K, and proposes a dynamic value and policy iteration methods that sequentially disaggregates states. Numerical results compare the approach with standard solvers from the literature.

**Ethical Considerations:**

(5) Excellent: The paper comprehensively addresses all of the applicable ethical considerations

**Nomination For Best Paper:**

No

**Questions For Authors:**

Please kindly clarify the connection with the existing work.

**Reproducibility:**

2: Some details are missing, but the paper still appears to be replicable with some effort.

**Strengths Of The Paper:**

+ Nice formalism of the state-abstraction MDP for infinite-horizon settings.
+ Strong initial numerical results.

The paper is very well written and easy to follow. I am appreciative of the formalism that the authors introduced to define their operators and the rigorous statements and proofs. The disaggregation scheme is also nicely designed and intuitive, specifically partitioning regions based on the partition error (Theorem 1). Finally, the results for the methodologies and benchmarks tested suggest that the approach is very promising

**Weaknesses Of The Paper:**

- Unclear about novelty and relationship to literature.

The study of state aggregation techniques for DP spans decades and I am still unclear what exactly are the novelties of this approach. In particular, the paper really lacks a more comprehensive analysis of existing methods and how they relate to their proposed policy and value iteration, as all concepts (include the projection) do closely relate to existing work. For example, the paper shares some relationship with some recent stream of state abstraction and options, aggregate-disaggregate methods, and even more classical literature on DPs such as the seminal work by Bean et al. (1987):

- Ciosek, Kamil, and David Silver. "Value iteration with options and state aggregation." arXiv preprint arXiv:1501.03959 (2015).
- Abel, David, et al. "Value preserving state-action abstractions." International Conference on Artificial Intelligence and Statistics. PMLR, 2020.
- Bean, James C., John R. Birge, and Robert L. Smith. "Aggregation in dynamic programming." Operations Research 35.2 (1987): 215-220.
- Schweitzer, Paul J., Martin L. Puterman, and Kyle W. Kindle. "Iterative aggregation-disaggregation procedures for discounted semi-Markov reward processes." Operations Research 33.3 (1985): 589-605.

These are just a few examples. Even the work by Chen et al. 2022, while cited, has not been developed further in the text to establish its relationship with the proposed method.



* Minor note:
- The text is well written but several letters are not capitalized correctly throughout; e.g., optimal Error Bound --> Optimal.

---

> ### Author Rebuttal · Authors · 2024-01-27
>
> Dear Reviewer,
>
> We value your thorough examination of the paper we submitted and the bibliographical points. In particular, we acknowledge the necessity to position and to demonstrate the novelty of our approach with a dedicated Related Work section.
>
> Aggregation techniques for optimization problems have been well-developed in the literature. Particularly, Starre, Loog, and Oliehoek (2022) offer a comprehensive overview that encompasses various aggregation processes, ranging from the metric-relative approach introduced by Abel et al. (2018) to the utilization of deep-learned representations. Gopalan, desJardins, Littman et al. (2017) also explores planning on fixed aggregation. Nevertheless, these methods lack strategies for building such abstractions without information on the MDP optimal solution.
>
> Multiple articles considered aggregation-disaggregation techniques to decompose optimization problems (Bean et al. 1987, Rogers et al. 1991). However, those methods are deterministic and therefore take no profit from the spatial structure of the problem, except for considering a graph-shaped problem.
>
> As you accurately pointed out, some works particularly address MDPs with aggregation-disaggregation processes, as exemplified by Schweitzer et al. (1985). The work from Schweitzer et al. almost immediately led to the analysis of Bertsekas \& Castañon (1988) we used in our comparison. They refine the aggregation process of Schweitzer et al. to take full advantage of the whole benefit in the Policy Iteration context.
>
> Recent works focusing on approximating MDPs often use options (actions lasting in time). For instance, Silver and Ciosek (2015) introduce options into a state abstraction to expedite planning, while Abel, Khetarpal et al. (2020) offer an overview of options tailored to specific classes of abstraction. Jothimurugan, Bastani, and Alur (2021) propose the identification of relevant subgoals as regions to accelerate planning. The temporal abstraction is an important alternative hypothesis that can bring a deeper understanding of MDP dynamics, but it induces more computations.
>
> There are few works that build efficient numerical aggregation-disaggregation. What we propose not only accelerated the exact solving process but also provided an effective way to build abstraction with a limited approximation, which has not been done before as most recent papers only study the resulting approximation following a given aggregation rule.
>
> Sincerely,
>
> The authors

---

### Official Review · Reviewer_DDQW · 2024-01-22

**Significance And Importance:** 3
**Soundness:** 4
**Novelty:** 3
**Clarity:** 4
**Overall Evaluation:** 2
**Confidence:** 3

**Weaknesses:**

1: Minor weaknesses that are easily fixable.

**Contributions Of The Paper:**

As stated in its title, this paper proposes a progressive state disaggregation method for (infinite horizon) dynamic programming.

After discussing background on state abstraction/aggregation, it first defines a state aggregation process such that the state weights within a region sum to one, which induces an abstract MDP.  The authors also introduce (in parallel) an approximation of the value function that is piecewise constant (per region of the aggregation), which induces a projected optimal Bellman operator.

The next section shows that the fixed point solutions of the abstract MDP and of the approximate dynamic programming are equivalent, and that the projected Bellman operators at hand converge (before discussing their complexity).

Then, after bounding the error between a piecewise constant value function and the optimal value function, algorithms are presented (based on VI, QVI and PI) that progressively refine an initially very coarse state aggregation and stop when some error bound is guaranteed.
The complexity of these algorithms is discussed, showing that, in the worst case (which can provably happen), they may do worse than standard variants.  Experiments illustrate the fact that, depending on the problem type, one may benefit or not from using this state aggregating algorithms.

**Ethical Considerations:**

(1) Not Applicable: The paper does not have any ethical considerations to address

**Nomination For Best Paper:**

No

**Questions For Authors:**

1. Is the abstract MDP derived from a state aggregation different from previous work? (e.g., from Abel et al.)

2. Definition 3: Do you confirm that $\cal T$ should not be in $\cal T$?

3. In PDPI, is the aggregation process restarted from scratch after each new policy improvement?

**Reproducibility:**

2: Some details are missing, but the paper still appears to be replicable with some effort.

**Strengths Of The Paper:**

As far as I know, this is one of the few works on state aggregation that proposes a sound method for aggregating states and ensuring convergence to within some distance to the original problem's optimal solution.

Overall the paper is well organized, well written and clear. It seems technically sound to me, most elements to understand the proofs are provided (even though not all proofs are detailed).

The experiments conducted seem sufficient to illustrate the main findings.

**Weaknesses Of The Paper:**

IMHO the main weakness is that no code is provided or promised.

I would also have been curious to see more of the algorithms' behavior, e.g., which value of $K$ is reached, in how many iterations, ... But there is not much space left.


The following is a list of small comments or issues, but not serious weaknesses:

- [Title] The authors may want to add a line break before "for" to better balance the title.

[39]
- "Van Dijk" -> "van Dijk"

[67]
<- "(Dean and Givan 1997) employs" (A parenthesis cannot act as a subject.)
-> "Dean and Givan (1997) employ"

[68]
<- "(Singh, Jaakkola, and Jordan 1994) introduces"
-> "Singh, Jaakkola, and Jordan (1994) introduce"
   [I don't list other issues of this type.]

[155]
<- "to the reward an transition"
-> "to the reward and transition"

[157]
<- ".One"
-> ". One"

[165]
<- "The practical solving of a MDP, can [...]"
-> "The practical solving of an MDP can [...]"

[173-174] Where are the subject and verb of this sentence?

- I find using "let" without "us" or "me" quite unusual. But maybe that's valid.
  I am more worried about the cases where there is no verb following "let". For instance, I would add "be" in several cases.

- The authors may want to make it clear whether their abstract MDP (derived from a state aggregation) is different from previous work.
  Among other things, this would help me understand if pathologies may happen as described by Tagorti et al. ( https://inria.hal.science/hal-00907315/ ).


[217]
<- "Those value function are"
-> "Those value functions are"

- Definition 3: $\cal T$ should not be in $\cal T$. That's an operator, not a value function.

[239+347]
<- "theirs complexity"
-> "their complexity"
or "their complexities" ??

[247]
<- "contains"
-> "contain"

[279]
<- "those $Q$-value"
-> "those $Q$-values"

[305]
<- "if the optimal"
-> "is the optimal"

[380]
<- "of theorem 1"
-> "of Theorem 1"

[Proof of Theorem 1] It would be nice to point out references about these two "classical inequalities".

[432]
<- "PDQVI provide"
-> "PDQVI provides"

- Formulas after line 469: The infinite norms should be replaced by absolute values.

[493]
<- "one consider"
-> "one considers"

[507]
<- "the Proposition 6"
-> "Proposition 6"

[516]
<- "the Figure 1"
-> "Figure 1"

[528]
<- "are necessary"
-> "is necessary"

[531]
<- "Value Iteration algorithm"
-> "the Value Iteration algorithm"

<- "Bertsekas Adaptive Aggregation"
-> "Bertsekas' Adaptive Aggregation"

<- "Chen Adaptive Aggregation"
-> "Chen's Adaptive Aggregation"

[591]
<- "for each queue which [...]"
-> "for each queue, which [...]"

[616]
<- "coefficient"
-> "coefficients"

[617]
<- "which implies"
-> ", which implies"

[625]
<- "Note that, the partition [...]"
-> "Note that the partition [...]"

[650]
<- "As disaggregation methods faces [...]"
-> "As disaggregation methods face [...]"

[References]
- Please protect capital letters in title when needed, e.g., for "Markov" or "MDP", but don't protect all capital letters either (e.g., line 700).

---

> ### Author Rebuttal · Authors · 2024-01-27
>
> Dear Reviewer,
>
> We value your thorough examination of the paper we submitted and the associated suggestions. Through this, we add the theoretical relation to the spatial abstraction classes of Abel, Hershkowitz \& Littman (2016) and add information about our algorithm outcome. We implemented the writing advice as well.
>
> 1) Our algorithm provides a final abstract MDP that checks multiple criterions. First, our two value-based algorithms PDVI and PDQVI provide concrete instances of abstractions that are part of the class $\phi_{Q^*, \varepsilon}$ described by Abel, Hershkowitz \& Littman (2016). Our algorithms gather states having respectively close optimal value and close optimal Q-value. Concerning the policy-based version of our method, the abstraction assembles states having close value under any policy explored by the algorithm. This results in a much thinner partition.
>
> 2) We admit that there is a typo in Definition 3. Tsitsiklis and Van Roy (1996) claim that the value function $\Pi \mathcal{T}^* V$ is the best piecewise constant approximation of the updated value function $\mathcal{T}^* V$ which can be expressed as $$\forall V \in \mathbb{R}^{\mathcal{S}}, ~\Pi \mathcal{T}^* V \in \operatorname{argmin}_{V' \in \mathcal{P}} \| V' - \mathcal{T}^* V \|_2,$$ defining $\mathcal{P}$ as the set of piece-wise constant value function.
>
> 3) The number of regions highly depends on the model and on the final precision. Considering the Four Rooms model (Hengst, Hierarchical Approaches, 2012) solved with a precision of $10^{-3}$. The value-based algorithm can gather S=200 states into K=20 regions (10 states by region on average). Each disaggregation step finds 1 or 2 new regions, which result into 15 disaggregation steps. The policy-based method uses a much thinner aggregation (2 or 3 states instead of 10), which is not reset during policy improvement. However, dividing by 3 the number of states can reduce the transition matrix size up to a factor 9 while not losing any information about the MDP.
>
> As suggested, we will add the work from Tagorti et al. (2013) that demonstrates that there always exists an aggregation refinement that worsens the distance to optimal value function. We avoid such problems by wisely choosing the next aggregation refinement to effectively improve the current value function.
>
> The code is available at https://github.com/ArticleAnonymous/ICAPSagg where we added new MDP instances.
>
> Sincerely,
>
> The authors

---

### Official Review · Reviewer_NgJU · 2024-01-23

**Significance And Importance:** 2
**Soundness:** 3
**Novelty:** 3
**Clarity:** 3
**Overall Evaluation:** 1
**Confidence:** 3

**Weaknesses:**

0: Minor weaknesses requiring some work to be addressed for the paper to be accepted.

**Contributions Of The Paper:**

This paper presents an approach for applying dynamic programming model-based reinforcement learning (RL) to environments with large state space. The approach is based on state space abstraction. In particular, a new state abstraction method is proposed, which starts considering a single state and progressively disaggregates it generating a set of state partitions that can be considered as single states introducing a small error. The original contribution is the introduction of state disaggregation during value function iteration. The disaggregation is applied to three standard algorithms obtaining related improved methods, namely, Progressive Disaggregation Value Iteration (PDVI), Progressive Disaggregation Q-Value Iteration (PDQVI), and Progressive Disaggregation Policy Iteration (PDPI). A theoretical derivation and an empirical evaluation of the three algorithms are reported, showing that the proposed methods are well-founded and they can outperform both standard dynamic programming algorithms (i.e., Value Iteration and Modified Policy Iteration) and state-of-the-art state aggregation methods (i.e., Bertsekas Adaptive Aggregation and Chen Adaptive Aggregation).

**Ethical Considerations:**

(1) Not Applicable: The paper does not have any ethical considerations to address

**Nomination For Best Paper:**

No

**Questions For Authors:**

See section "Weaknesses Of The Paper".

**Reproducibility:**

2: Some details are missing, but the paper still appears to be replicable with some effort.

**Strengths Of The Paper:**

- The theoretical derivation of the proposed method seems to be well-defined.
- The writing quality is good. Despite the complexity of the topic, I managed to follow the main passages of the presentation.
- Mathematical notation is precise and it seems to be correct. The technique seems to be technically sound
- The topic, namely, model-based RL with MDPs, is very relevant to ICAPS
- Although the work looks a bit "old-fashion" I like the precise (and also quite clear) theoretical derivation.

**Weaknesses Of The Paper:**

- The comparison between the proposed Progressive Disaggregation methods and the state-of-the-art Adaptive Aggregation methods (i.e., Bertsekas Adaptive Aggregation and Chen Adaptive Aggregation) should be made more explicit to allow the reader to better and simply understand what is the methodological difference that allows Progressive Disaggregation to achieve a performance improvement. This comparison could be put in a specific section or in a related work section.
- The related work section is missing, I suggest introducing it and using it to show the differences between SOTA methods and the proposed method. This would allow the reader to better understand which is the original contribution of the paper, which is currently not stressed enough, from my point of view.
- The contribution seems original but the literature cited in the paper is not recent, hence the topic seems not to be very hot. The only recent work cited is (Chen et al. 2022), a workshop paper presenting an Adaptive State Aggregation Algorithm for Markov Decision Processes to which the proposed method has been compared.
- In analyzing the literature, maybe authors could also consider point-based policy iteration and value iteration methods (and related literature). They have been introduced in the context of POMDPs but I think the goal is similar and a comparison could be interesting. Furthermore, this analysis could allow the authors to extend the bibliography which is at the moment quite short (and it considers not very recent works).
- Since the recent literature about RL for high dimensionality environments is very focussed on model-free methods and model-based methods that learn the model, I think this work should make more explicity the relationship with that part of the literature to better explain the positioning compared to the rest of the literature. I think this would strenghten the paper and make it more interesting also for researchers working with recent model-free and model-based RL methods.
- As in the previous point, I suggest making explicit the relationship between the proposed work and model-approximation/policy-approximation/policy-gradient/DRL methods. Why and when should a reader use the proposed approach instead of recent/popular RL methods on high-dimensionality domains?
- Again, a comparison between this approach and online approaches, such as Monte Carlo Tree Search based methods should be considered. Why and when should a reader use the proposed approach instead MCTS-based RL methods on high-dimensionality domains?
- It would be nice to make available the code to guarantee reproducibility

---

> ### Author Rebuttal · Authors · 2024-01-27
>
> Dear Reviewer,
>
> We appreciate your thorough examination of the paper we submitted and the associated suggestions. Through this, we acknowledge the necessity of emphasizing the paper's positioning within State Abstraction and the broader context of Reinforcement Learning in a dedicated 'Related Works' section. We have added here references necessary to situate our work.
>
> Recent advancements in abstraction have improved the comprehension of underlying structures in MDPs. Our work focuses on spatial abstraction, not only solving the exact process using aggregation but also introducing a novel method for building abstractions with bounded error which is a complex combinatorial challenge for large MDPs. This approach stands out from recent papers, which typically consider aggregation based on predefined rules.
>
> Our main contribution is to create abstractions that are used to estimate a solution of the original MDP. We selected papers as Bertsekas \& Castañon (1988) and Chen et al. (2022) which also provide a way to aggregate states accelerating convergence. Those two techniques do not take advantage of aggregation to describe the spatial structure of the MDP while our method keeps trace of previous aggregations while building the final abstraction.
>
> Regarding spatial aggregation in Model-Based RL, Starre, Loog and Oliehoek's survey (2022) offers a comprehensive overview, ranging from metric-relative methods as proposed by Abel et al. (2018) to deep learned representations. Ferrer-Mestres (2020) also explores metric abstraction with a fixed number of regions. However, these approaches require solving a problem using knowledge unavailable before solving the original MDP. Our work addresses this gap effectively by grouping states with similar optimal values and close trajectories through optimal Bellman operator iteration.
>
> Considering POMDP, Point-based Value Iteration (Pineau, Gordon \& Thrun, 2003) use one state to represent a region and could indeed be added to the comparison. MCTS rely on a model-based local policy optimization, but will suffer from a state abstraction that loose the tree structure of the actions. Moreover, most policy-based approaches are geared towards Deep Learning methods using policy gradient. They are efficient in practice, but rarely ensure guarantees on quadratic or sup-norm error.
>
> The code is available at https://github.com/ArticleAnonymous/ICAPSagg where we added new MDP instances.
>
> Sincerely,
>
> The authors

---

### Meta-Review · Area_Chair_ECr3 · 2024-02-04

**Recommendation:** Accept (Poster)
**Confidence:** 3

**Metareview:**

In this paper, the authors work on the "classical" (i.e. non DL-based) methods for solving MDPs. The contributions are sound and interesting to the ICAPS community. The main issue with this paper is that the author did not fully position their work w.r.t. prior work on DP for RL that also used state aggregation techniques. If the paper is accepted I urge the authors to include a more complete discussion of this related work in their paper.

We appreciate your rebuttal responses!

**Ethical Considerations:**

(1) Not Applicable: The paper does not have any ethical considerations to address